# Partial-, Double-Enzymatic Dephosphorylation and EndoGluC Hydrolysis as an Original Approach to Enhancing Identification of Casein Phosphopeptides (CPPs) by Mass Spectrometry

**DOI:** 10.3390/foods10092134

**Published:** 2021-09-09

**Authors:** Barbara Deracinois, Aurélie Matéos, Audrey Romelard, Audrey Boulier, Julie Auger, Alain Baniel, Rozenn Ravallec, Christophe Flahaut

**Affiliations:** 1UMR Transfrontalière BioEcoAgro N° 1158, Univ. Lille, INRAe, Univ. Liège, UPJV, YNCREA, Univ. Artois, Univ. Littoral Côte d’Opale, ICV—Institut Charles Viollette, 59000 Lille, France; barbara.deracinois@univ-lille.fr (B.D.); aurelie.mateos@univ-artois.fr (A.M.); rozenn.ravallec@univ-lille.fr (R.R.); 2Ingredia S.A., 51 Av. Lobbedez-CS 60946, CEDEX, 62033 Arras, France; a.romelard@ingredia.com (A.R.); a.boulier@ingredia.com (A.B.); j.auger@ingredia.com (J.A.); a.baniel@ingredia.com (A.B.)

**Keywords:** casein phosphopeptides, endoGluC hydrolysis, enzymatic dephosphorylation, mass spectrometry

## Abstract

The identification of phosphopeptides is currently a challenge when they are part of a complex matrix of peptides, such as a milk protein enzymatic hydrolysate. This challenge increases with both the number of phosphorylation sites on the phosphopeptides and their amino acid length. Here, this paper reports a four-phase strategy from an enzymatic casein hydrolysate before a mass spectrometry analysis in order to enhance the identification of phosphopeptides and phosphosites: (i) the control protein hydrolysate, (ii) a two-step enzymatic dephosphorylation of the latter, allowing for the almost total dephosphorylation of peptides, (iii) a one-step enzymatic dephosphorylation, allowing for the partial dephosphorylation of the peptides and (iv) an additional endoGluC enzymatic hydrolysis, allowing for the cleavage of long-size peptides into shorter ones. The reverse-phase high-pressure liquid chromatography–tandem mass spectrometry (RP-HPLC-MS/MS) analyses of hydrolysates that underwent this four-phase strategy allowed for the identification of 28 phosphorylation sites (90%) out of the 31 referenced in UniprotKB/Swiss-Prot (1 June 2021), compared to 17 sites (54%) without the latter. The alpha-S2 casein phosphosites, referenced by their similarity in the UniProt database, were experimentally identified, whereas pSer148, pThr166 and pSer187 from a multiphosphorylated long-size kappa-casein were not. Data are available via ProteomeXchange with identifier PXD027132.

## 1. Introduction

Milk is, above all, the main food of the newborn in mammals and mandatory for its good growth, therefore containing all of the nutrients necessary, such as lipids, carbohydrates, vitamins and proteins. Nevertheless, in human food, milk holds a particularly important place beyond even infant food because of its richness in nutritional compounds of good qualities. Milk proteins made up of 20% whey proteins and 80% caseins are used by the dairy industry for their nutritional and techno-functional properties. These proteins are also a source of multiple molecules with biological activity that can be generated during their hydrolysis.

Among these molecules, casein phosphopeptides (CPPs) produced by the hydrolysis of αs1-, αs2-, β- and κ-casein are of relative interest [1]. Indeed, the caseins are phosphorylated proteins with a specific cluster of phosphorylation (pSer-pSer-pSer-Glu-Glu) that can bind several minerals, such as calcium, iron, zinc and magnesium [2,3,4]. This binding improves the solubility of minerals in the gastrointestinal tract and enhances their crossing through the intestinal barrier (enhancing their bioavailability). CPPs exhibit antioxidant properties by (i) the scavenging of free radicals, (ii) chelating the pro-oxidant metallic ions and (iii) the scavenging of reactive oxygen species (ROS), as highlighted by several in vitro studies [5,6,7,8]. CPPs are also recognized as acting as an anticariogenic agent by both the inhibition of odontopathogenic bacteria, such as Streptococcus mutans, and by a supply of calcium and phosphate through the formation of nanoclusters at the surface of tooth enamel [9,10]. Bone mineralisation is promoted by the CPP’s ability to bind calcium ions and to conserve their bioavailability during the gastro-intestinal digestion. Therefore, it exists as a great interest in the development of functional food and ingredients with a supply of CPPs that may have beneficial effects on human health. Indeed, the CPPs are well-known in Asia and especially in China, which represents more than 80% of new products launched with CPPs between 2015–2020 (Innova database).

Those products are launched for targeted populations who need to have the right calcium intake for all age categories combined. For example, products such as infant formula for bone growth are dedicated to babies. In the packaging, casein phosphopeptide labelling is used to reinforce the communication around the bone health properties of the finished product. In China the CPP dosage is regulated as 300 mg/100 g of powder of infant formula. The other targeted population are seniors, with different products proposed in the market: food supplements (tablets, sachets and capsules), but also functional foods, such as milk or yoghurts with added CPPs and calcium. With CPPs, the added calcium will be more bioavailable; this property is interesting in preventing bone loss for elderly people.

CPPs are generally supplied in the form of casein hydrolysates obtained by the enzymatic proteolysis of micellar caseins and, so, are supplied in a mixture with a lot of non-phosphorylated peptides. Among all milk caseins, 31 phosphorylation sites are referenced in UniprotKb. The in silico trypsic hydrolysis (PeptideCutter, https://web.expasy.org/peptide_cutter/, accessed on 1 June 2021) of bovine milk caseins (without missed cleavages) lists five mono-phosphorylated CPPs (αS1-casein 52–57: pS56, 121–134: pS130; αS2-casein 40–47: pS46, 153–164: pS158; β-casein 48–63: pS50), one single doubly-phosphorylated CPP (αS1-casein 58–73: pS61–pS63), one single tri-phosphorylated CPP (αS2-casein 141–151: pS144–pS146–pS150), four tetra-phosphorylated CPPs (αS2-casein 17–36: pS23–pS24–pS25–pS28, 61–85: pS71–pS72–pS73–pS76; β-casein 17–39: pS30–pS32–pS33–pS34; κ-casein 138–189: pS148–pT166–pS170–pS186) and a single penta-phosphorylated CPP (αS1-casein 74–94: pS79–pS81–pS82–pS83–pS90).

Various methods for the mass spectrometry-based detection of CPPs have been developed over the years, with different approaches toward the, step of enrichment of CPPs, ranging from the selective precipitation of CPPs with CaCl_2_ [11,12] to the concentration of CPPs using metal ions (TiO_2_, Ti^4+^, Fe^3+^, Ga^3+^) known as immobilized metal ion affinity (IMAC) [13,14,15,16], strong cation exchange chromatography [17], polymer-based metal–ion affinity capture (PolyMAC) and metal oxide chromatography (MOC) [18,19]. However, among the large theoretical number of CPPs that can be generated by hydrolysis, only a small part (mainly the monophosphorylated CPPs) could be successfully identified by the use of the aforementioned pre-purification steps combined with the proteomic approach based on reverse-phase high-pressure liquid chromatography combined with tandem mass spectrometry (RP-HPLC-MS/MS), the database and bioinformatics [20]. These methods do not allow for the identification of multi-phosphorylated and/or large peptides, and the identification of the phosphorylation site occupancy is, finally, always under-evaluated. Indeed, multi-phosphorylated peptides generally have a low ionization efficiency in the positive ion detection mode and are difficult to identify by low-energy collision-induced dissociation (CID) because both the labile phosphate group often generates dominant peaks based on the neutral loss of H_3_PO_4_ (98 Da) and the b- and y-ions are hardly detected in CID spectra. Concomitantly, the pioneer work [21,22] of the chemical modification of phosphate groups using a beta-elimination reaction combined to the derivatization reaction based on the Michael addition generated numerous published works using different functional chemical groups.

The aim of the present study was the development of a relatively efficient strategy (Figure 1), without the phosphopeptide enrichment step or chemical dephosphorylation- and derivatization steps, in order to detect, in a mixture of peptides and phosphopeptides, both peptides and, above all, a maximum of CPPs, including multiphosphorylated and large-size CPPs, by using RP-HPLC, high-resolution mass spectrometry (MS) and bioinformatics. Our goal was not to obtain an exhaustive view of phosphorylation site locations of bovine caseins or to map new phosphorylation sites, but to set up a routine analytical method to quickly and efficiently evaluate the batch-to-batch phosphopeptide content of industrial enzymatic hydrolysates of micellar caseins. An industrial commercial casein hydrolysate was used as a model hydrolysate for this purpose. This approach consisted of the us of a double- and a partial-enzymatic dephosphorylation of CPPs, as well as an enzymatic hydrolysis using an endoGluC protease, before peptidomics analysis (Figure 1).

## 2. Material and Methods

### 2.1. Materials

Two batches of CPP hydrolysates were prepared by the Ingredia S.A. manufacture (St-Pol-Sur-Ternoise, France) using an industrial process. Briefly, micellar caseins (ratio micellar caseins/whey proteins (92:8)) were hydrolyzed by a food-grade trypsin and dried by atomization using the Mini Spray Dryer B-290 from BUCHI (Rungis, France). The enzyme Glu-C endoproteinase sequencing grade (EC 3.4.21.19) from *Staphylococcus aureus V8* was obtained from Promega (Madison, WI, USA). Alkaline phosphatase (EC 3.1.3.1) from calf intestine and acid phosphatase (EC 3.1.3.2) from potato were purchased from Sigma-Aldrich (Merck KGaA, Darmstadt, Germany). All other reagents were of mass spectrometry-grade.

### 2.2. Preparation of Control CPP Hydrolysates

Dried CPP hydrolysates were solubilized at a concentration of 1 mg/mL in a solution of LC-MS-grade water acidified with 0.1% trifluoroacetic acid (TFA), and centrifuged for 10 min at 8000× *g*. Pellets were removed and supernatants were called “control CPP hydrolysates” (control CPPs) and submitted to peptidomics analysis.

### 2.3. Preparation of Doubly Dephosphorylated CPP Hydrolysates

A 2-step method of dephosphorylation was applied to CPP hydrolysates. The protocol used was adapted from Matéos et al. [23]. CPP hydrolysates were solubilized at a concentration of 3 mg/mL in Tris-HCl 20 mM pH 8.5, 0.11 mM MgCl_2_ and firstly treated with alkaline phosphatase (EC 3.1.3.1) at 6.1 U/mL during 20 h at 37 °C under weak agitation. However, this protocol was slightly improved by (i) the use of protease inhibitors (cOmplete^TM^ EDTA-free, Roche Diagnostic, Darmstadt, Germany), (ii) the addition of 0.11 mM ZnCl_2_ instead of Zn(CH_3_COO)_2_ and (iii) no dialyze but a drying of the dephosphorylated hydrolysates using a centrifugal evaporator at a temperature of 40 °C (miVac Centrifugal Vacuum Concentrators, Gene Vac, Ipswich, UK). In a second step, the aforementioned hydrolysates generated by alkaline dephosphorylation were solubilized in 50 mM sodium citrate and pH 5.8 at 3 mg/mL, and subjected to a 5 h-treatment at 30 °C with acid phosphatase (EC 3.1.3.2) at 0.012 U/mL at pH 6.0 under weak agitation, as mentioned in Matéos et al. [23]. As for the first dephosphorylation step, some minor improvements were made (addition of protease inhibitors, incubation time (5 h instead of 24 h) and drying by centrifugal evaporation). Dried peptides were dissolved in H_2_O, 0.1% TFA to obtain a CPP hydrolysates concentration of 1 mg/mL and centrifuged for 10 min at 8000× *g*. Pellets were removed and supernatants were called “doubly-dephosphorylated CPP hydrolysates” (DD CPPs) and submitted to peptidomics analysis.

### 2.4. Preparation of Partially Dephosphorylated CPP Hydrolysates

Partial dephosphorylation of the CPPs was also carried out on CPP hydrolysates. For this purpose, a single dephosphorylation using the alkaline phosphatase (EC 3.1.3.1) was carried out according to the aforementioned protocol, but for a period of time limited to 1 h. Dried peptides were dissolved in H_2_O, 0.1% TFA to obtain a theoretical CPP concentration of 1 mg/mL, and centrifuged for 10 min at 8000× *g*. Pellets were removed and supernatants were called “partially-dephosphorylated CPP hydrolysates” (PD CPPs).

### 2.5. Preparation of endoGluC Digested CPP Hydrolysates

CPP hydrolysates were also subjected to enzymatic hydrolysis by Glu-C endoproteinase (EC 3.4.21.19). CPP hydrolysates were dissolved at 1 mg/mL in hydrolysis buffer (ammonium bicarbonate 100 mM, pH 7.8) and enzymatic hydrolysis was carried out for 16 h at 37 °C with an enzyme:substrate (E/S) ratio of 1:100. After centrifugation for 10 min at 8000× *g*, pellets were removed and supernatants were called “endoGluC-digested CPP hydrolysates” (endoGluC CPPs).

### 2.6. CPP Identification by RP-HPLC-MS/MS and Database Search

Ten µL of control-, DD-, PD- and endoGluC-CPPs were chromatographically separated on an ACQUITY UPLC system (Waters Corporation) using a HALO^®^ AQ-C18 column (150 × 2.1 mm, 2.7 µm, Advanced Materials Technology GmBH, Wilmington, NC, USA). Eluent A was H_2_O MQ containing formic acid (0.1%, *v*/*v*) and eluent B was acetonitrile (ACN) containing formic acid (0.1%, *v*/*v*). The ACN gradient (flow rate 0.5 mL/min) was as follows: from 5% to 15% eluent B over 25 min, from 15% to 30% over 20 min, from 30% to 50% over 5 min and then 5 min 95% eluent B. The eluate was directed into the electrospray ionization source of the Q-TOF Synapt G2-Si™ (Waters Corporation, Manchester, UK). MS analysis was performed in sensitivity, positive ion and data-dependent analysis (DDA) modes. The source temperature was set to 150 °C and the capillary and cone voltages were set to 3000 and 60 V, respectively. MS data were collected for *m*/*z* values in the range of 100 Da and 2000 Da, with a scan time of 0.2 s and a lock mass correction of 556.632 *m*/*z*, corresponding to simply charged leucine enkephalin. A maximum of 10 precursor ions were chosen for MS/MS analysis, with an intensity threshold of 100,000. MS/MS data were collected using the collision-induced dissociation (CID) mode and a scan time of 0.1 s at an energy collision of 8 V to 9 V for low *m*/*z* and at a range of 40 V to 90 V for high *m*/*z*. All peptidomic analyses were performed in duplicates.

Database searches were performed via the PEAKS Studio 8.5 software (Bioinformatics Solutions Inc., Waterloo, ON, Canada) using the UniProtKB/Swiss-Prot databases restricted to *Bos taurus* organism (10 September 2018). A mass tolerance of 35 ppm and an MS/MS tolerance of 0.2 Da were allowed. The data searches were performed, notifying the choice of trypsin as enzyme, and three missed cleavage sites were allowed. Variable serine-, threonine- and tyrosine-phosphorylations and methionine oxidation were also considered, with a maximum of 7 post-translational modifications allowed by peptide. The relevance of (phospho)peptide identities was judged according to their identification score, which was returned by PEAKS Studio 8.5 using a *p*-value of 0.05 (*p* < 0.05) and a false discovery rate (FDR) < 1%.

Moreover, in order to envision peptide abundance, the peptide identification data were exported from PEAKS Studio 8.5 to a home-built Microsoft Excel sheet to generate heat maps of the amino acid occurrences (the number of times the amino acid was found in an identified peptide and the number of times the phosphorylation was detected) for each major protein, according to control-, DD-, PD- and endoGluC-CPP hydrolysates. GraphPad PRISM 6.01 (San Diego, CA, USA) was used to draw the amino acid occurrence heat maps displayed in the figures (where the left *y*-axis represents the amino acid occurrence, the right *y*-axis represents the phosphorylation occurrence and the *x*-axis represents the amino acid sequence of the protein).

The mass spectrometry proteomics data have been deposited to the ProteomeXchange Consortium via the PRIDE [24] partner repository with the dataset identifier PXD027132. 

## 3. Results and Discussion

### 3.1. MS Data Comparison of the Four Phases of the Protocol

#### 3.1.1. MS Scans, MS/MS Scans and Identified Peptides

Due to the data-dependent analysis mode used to manage the MS and MS/MS data acquisition, the number of MS and MS/MS scans provides a rapid overview of the peptide heterogeneity of the different samples analyzed. Indeed, the time spent to acquire MS data alters the time remaining to acquire MS/MS data, since the time scan is fixed and invariable [25,26]. In other word, a high number of MS scans, which denotes a low quantity of ions displaying a correct intensity to be fragmented, is associated with a low number of MS/MS scans, since few ions have been selected in MS mode and fragmented. Conversely, a low number of MS scans, which denotes a high number of ions with an intensity above the threshold defined for fragmentation, is associated with a high number of MS/MS scans, since numerous ions have been selected in MS mode and fragmented. As illustrated in Figure 2a, on average, 10,300 (±150) MS scans and 2500 (±213) MS/MS scans were recorded, regardless of the hydrolysates analyzed. Two-way ANOVA multiple comparisons using a *p*-value inferior to 0.05 (*p* < 0.05) were carried out from the MS data and performed in duplicates. Figure 2a reveals statistically significant differences, as highlighted by the lowercase letters, between the MS and MS/MS scan numbers related to DD-CPPs, endoGluC-CPPs and the two others conditions. The average number of MS scans registered during the HPLC-MS/MS runs of DD- and endoGluC-CPP hydrolysates is significantly lower, whereas the number of MS/MS scans registered during these runs is higher compared to the control- and PD-CPP hydrolysates. The fact that a lower number of MS scans and a higher number of MS/MS scans are registered during the RP-HPLC-MS/MS analyses of DD- and endoGluC-CPP hydrolysates suggests that the number, the intensity or both of the ions chosen for fragmentation in these samples are more important than in the other.

The industrial trypsin hydrolysis of milk micellar caseins generates a high number of peptides and CPPs due to the limited cleavage specificity of the food-grade trypsin used. Therefore, and as expected, the supplemental use of analytical enzymatic protocols dedicated to a better detection of CPPs change the detected number of peptides and CPPs in the DD-, PD- and endoGluC-CPP hydrolysates. Thus, as illustrated in Figure 2b (peptides), the number of identified bovine peptides is significantly different (lowercase letters) between control- and PD-/endoGluC-CPP hydrolysates, but also between DD- and endoGluC-CPP hydrolysates. The endoGluC hydrolysis of CPP hydrolysates yields around 27% more peptide identifications, with almost 300 (305) identified peptides compared to 240 identified peptides from the control-CPP hydrolysate. Obviously, the simplest explanation relies on the endoGluC-proteolytic cleavage of large peptides, unidentifiable with the MS method used, which became identifiable as smaller peptides [27]. In the same way, but to a lesser extent, partially enzymatic dephosphorylation enhances the peptide identifications too, with 19% (in average) more identified peptides from the PD-CPP hydrolysate compared to the control-CPP hydrolysate. It is worth noting that, despite a reduction in the peptide heterogeneity as a result of the enzymatic double dephosphorylation, the number of identified peptides from the DD-CPP hydrolysate is, as expected, not statistically different when compared to the number of peptides identified from the control-CPP hydrolysate.

#### 3.1.2. CPP Identification

Concomitantly, the analysis of the number of identified CPPs is instructive. Indeed, an average of 58, 25, 101 and 27 CPPs were identified in the control-, DD-, PD- and endoGluC-CPP hydrolysates, respectively. Thus, with an average of 25 CPPs identified, and, as previously reported in [23], the double enzymatic dephosphorylation obviously does not allow for a complete dephosphorylation. By contrast, chemical dephosphorylation by beta-elimination is recognized as complete. However, the combination to the Michael addition causes the α-carbon racemization of Ser and Thr, and the formed diastereomers can be separated by LC, resulting in a decrease in sensitivity and an increase in complexity [28,29]. Moreover, the double dephosphorylation leads to a non-significant decrease of the number of identified CPPs (Figure 2b, CPPs). The same tendency is also observed for the endoGluC-CPP hydrolysate, with the exception that the number of total identified peptides is significantly higher than the control-CPP hydrolysate.

Conversely, the number of identified CPPs is the highest for the PD-CPP hydrolysate and is significantly higher compared to the three others. Combined with the total number of identified peptides from the PD-CPP hydrolysate, these data suggest that the supplemental identified peptides correspond to identifications of CPPs.

### 3.2. CPPs and Phosphosites

In silico tryptic hydrolysis of bovine caseins alpha-S1- (CASA1), alpha-S2- (CASA2), beta- (CASB) and kappa-casein (CASK) yields 12 tryptic distinct CPPs with 31 different phosphorylation sites when no missed cleavage was mentioned (Figure 3, gray rectangles and gray arrows, respectively). These 12 CPPs are distributed as short- and long-size peptides, and monophosphorylated to pentaphosphorylated peptides, as follows (the code used is normal case, number of amino acids and superscript, number of phosphorylation site): four peptides for CASA1 (6^1^, 14^1^, 16^2^ and 21^5^), five peptides for CASA2 (8^1^, 11^3^, 12^1^, 20^4^ and 25^4^), two peptides for CASB (16^1^ and 23^4^) and one long-size peptide for CASK (52^4^). However, the usual RP-HPLC-MS/MS analysis of an industrial tryptic hydrolysate of caseins, i.e., control sample, leads to the identification of 35 CPPs for 17 phosphorylation sites, corresponding to the different cleavages of 8 (out of 12) aforementioned in silico tryptic CPPs (Figure 3, black lines and black dots). Although αs1-, αs2-, β- and κ-casein are all phosphoproteins, they vary strikingly in their degree of phosphorylation [30]. Therefore, the complexity of multiphosphorylated CPPs increases with the number of phophorylation isoforms [30], thus multiplying the CPP number. Moreover, the low cleavage specificity of the industrial trypsin enzyme, which, in turn, increases the diversity of the amino acid backbone of CPPs, leads to the experimental identification of a much higher number of CPPs than expected in silico, and brings the total number of distinct CPPs, experimentally identified in the control-CPP hydrolysate, to 89.

The peptidomic approaches of the control-CPP- and endoGluC-CPP hydrolysates leads to the identification of 89 and 46 CPPs, corresponding to 8 and 10 in silico-predicted tryptic CPPs and 17 and 16 phosphorylated sites, respectively. Concomitantly, those performed from DD-CPP- and PD-CPP hydrolysates identify 36 and 149 CPPs derived from 3 and 10 in silico-predicted tryptic CPPs with 8 and 25 phosphorylated sites, respectively. These discrepancies regarding (i) the number of identified CPPs, (ii) the number of in silico-predicted tryptic CPPs and (iii) the number of occupied phosphorylation sites are illustrated in Figure 3 (red and blue lines and red and blue dots compared to black and dashed lines and black and dashed dots). Indeed, regardless of the casein, the blue and red lines and blue and red dots are always positioned above the black and black dashed lines and dots in the phosphorylation regions, except in the regions (second half of CASA1; and second in-silico-predicted tryptic peptide of CASB) where the number of phosphorylation sites is low. Overall, the peptidomic approach performed from the PD-CPP hydrolysate (blue line and blue dots) identifies more CPPs and more occupied phosphorylation sites in the multiphosphorylated protein regions. The peptidomic approach of the endoGluC-CPP hydrolysate identifies 46 CPPs derived from 10 in silico-predicted tryptic CPPs, which represent 16 phosphorylated sites (Figure 3, black dashed lines and dots). Interestingly, two in-silico-predicted CPPs (CASA2 40–47; 8^1^: pSer46 and CASK 138–189; 52^4^: pSer148, pThr166, pSer170, pSer187) were not identified from the control-, DD- and PD-CPP hydrolysates. However, the latter were identified from the endoGluC-CPP hydrolysate: the first one was identified as the in-silico-predicted monophosphorylated octapeptide (probably issued from a longer peptide resulting from a missed trypsin cleavage), whereas the second was identified as (i) one medium-size non phoshorylated peptide (CASK: 138–158) and (ii) one medium-size monophosphorylated peptide (CASK: 162–190, pSer170). Unfortunately, this strategy failed to detect and confirm the presence of the phosphate on the phosphosites pSer148, pThr166 and pSer187 of CASK. The Appendix A lists the 645 identified peptides, including 262 phosphopeptides, using this four-phase strategy.

Overall, taken together, the four hydrolyses (one industrial (control) and three analytical) lead to the identification of both CPPs derived from the 12 (out of 12) in-silico-predicted tryptic peptides and of 28 (out of 31) phosphorylated sites. Herein, we bring evidence of the phosphorylation of pSer144, pSer146 and pSer150 of CASA2. These pSer are currently referenced in the Uniprot database as “phosphorylated by similarity” based on a 1998 study related to the sequence analysis of *Camelus dromedarius* milk caseins [31]. The three missing phosphorylation sites are those present on the multiphosphorylated long-size CPP from CASK and corresponding to pSer148, pThr166 and pSer187. Besides, it is also important to note that, regardless of the hydrolysis method used, the phosphorylation degree of the multiphosphorylated long-size CPP from CASK still remains largely under-detected, as well as at the level of the amino acid backbone more than the phosphorylation sites. Indeed, concerning all articles devoted to the development of analytical methods to characterize phosphopeptides and phosphosites, CASK is never used as a phosphoprotein model. Until recently, a new chemical dephosphorylation/Michael addition-based method leading to the conversion, from multiple phosphorylated peptides of multiple phosphate groups into perfluoroalkyl groups, has been reported for the identification and the quantification of mono- and multi-phosphorylated peptides by a fluorous HPLC column coupled to mass spectrometry [32]. Although the non-coelution of peptides and fluorous-derivatized peptides is clearly evidenced, unfortunately this article does not discuss the monoisotopic profile of fluorous-derivatized peptides and does not report the MS/MS spectra of fluorous-derivatized peptides.

## 4. Conclusions

The diversity of peptides generated during enzymatic hydrolysis can easily be predicted with the in silico model [33,34]. However, according to the non-perfect cleavage specificity of the enzyme used, the real peptide heterogeneity of hydrolysates cannot be accurately predicted in silico. Moreover, the PTMs, such as the multiphosphorylation of proteins, enhance this peptide heterogeneity. Phosphorylated peptide identification is commonly managed using a conventional proteomic approach. However, the negative net charge of phosphate groups is a hurdle to the positive ionisation of peptides and results in a low positive ionization efficiency. The presence of several phosphate groups on one peptide amplifies this problem [20], and make the MS-based analysis of multiphosphorylated peptides challenging.

In this four-phase strategy, our goal was not to obtain an exhaustive view of the phosphorylation site locations of bovine caseins or to map new phosphorylation sites, but to set up a routine analytical method to quickly and efficiently evaluate the batch-to-batch phosphopeptide content of industrial enzymatic hydrolysates of micellar caseins. Herein, an almost total and a partial dephosphorylation step, combined with a secondary endoproteinase hydrolysis of long-size peptides, allowed for the detection and identification of more than 90% (28 out of 31 phosphosites) of the phosphorylated sites of CPPs, referenced in Uniprot. Only three phosphorylation sites, corresponding to pSer148, pThr166 and pSer187 from a multiphosphorylated long-size CASK CPP, were not identified. This innovative four-phase strategy does not require the use of a particular enrichment step or particular chromatographic column (a classical reverse phase is sufficient), nor any particular change in the RP-HPLC-MS/MS analysis parameters and data bioinformatics reprocessing. Concomitantly, its identification of non-phosphorylated peptides is efficient as well. Therefore, this strategy is applicable for both the analysis of phosphopeptides from the food industry hydrolysates and also for clinical phosphopeptidomic studies.

## Figures and Tables

**Figure 1 foods-10-02134-f001:**
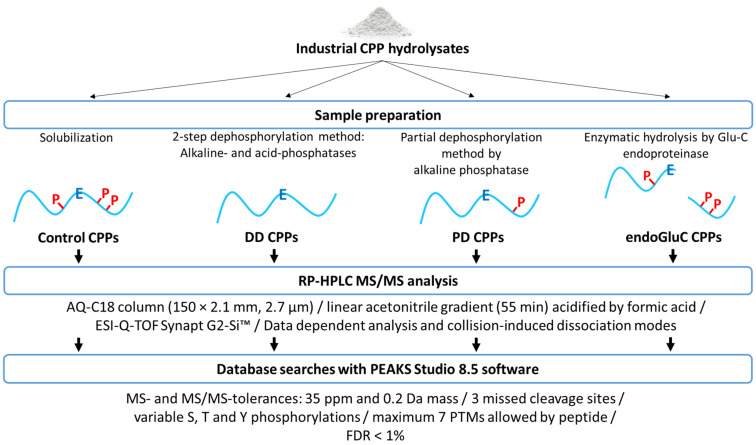
Schematic representation of the four-phase strategy to detect a maximum of CPPs, including multiphosphorylated and large-size CPPs, using RP-HPLC, high-resolution mass spectrometry (MS) and bioinformatics. Control—for no treatment; DD—for double-dephosphorylation; PD—for partial dephosphorylation; and endoGluC-CPPs—for endoGluC hydrolysis of CPP hydrolysates. PTMs, post-translational modifications; P, phosphosites; E, glutamic acid.

**Figure 2 foods-10-02134-f002:**
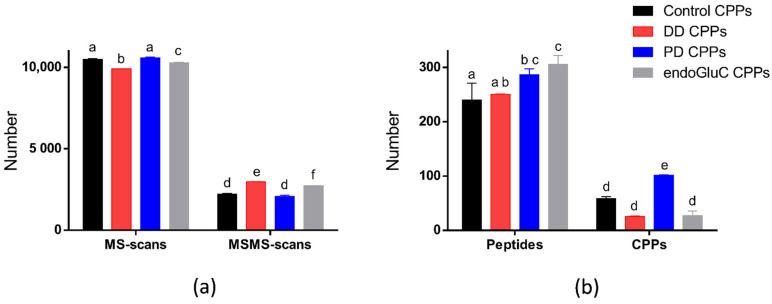
MS data, total identified peptides and identified casein phosphopeptides for the control- (in black), DD- (in red), PD- (in blue) and EndoGluC- (in gray) CPP hydrolysates. (**a**) Number of MS and MS/MSscans performed during the RP-HPLC-MS/MS runs; (**b**) number of identified peptides and identified CPPs with PEAKS Studio 8.5 software. All samples were analyzed in duplicate. The different lowercase letters report the statistically significant differences between data; an identical letter between two conditions indicates no statistically significant difference, whereas two different letters indicate a statistically significant difference (*p* < 0.05, 2-way ANOVA multiple comparisons). Control—for no treatment; DD—for double-dephosphorylation; PD—for partial dephosphorylation; and endoGluC-CPPs—for endoGluC hydrolysis of CPP hydrolysates.

**Figure 3 foods-10-02134-f003:**
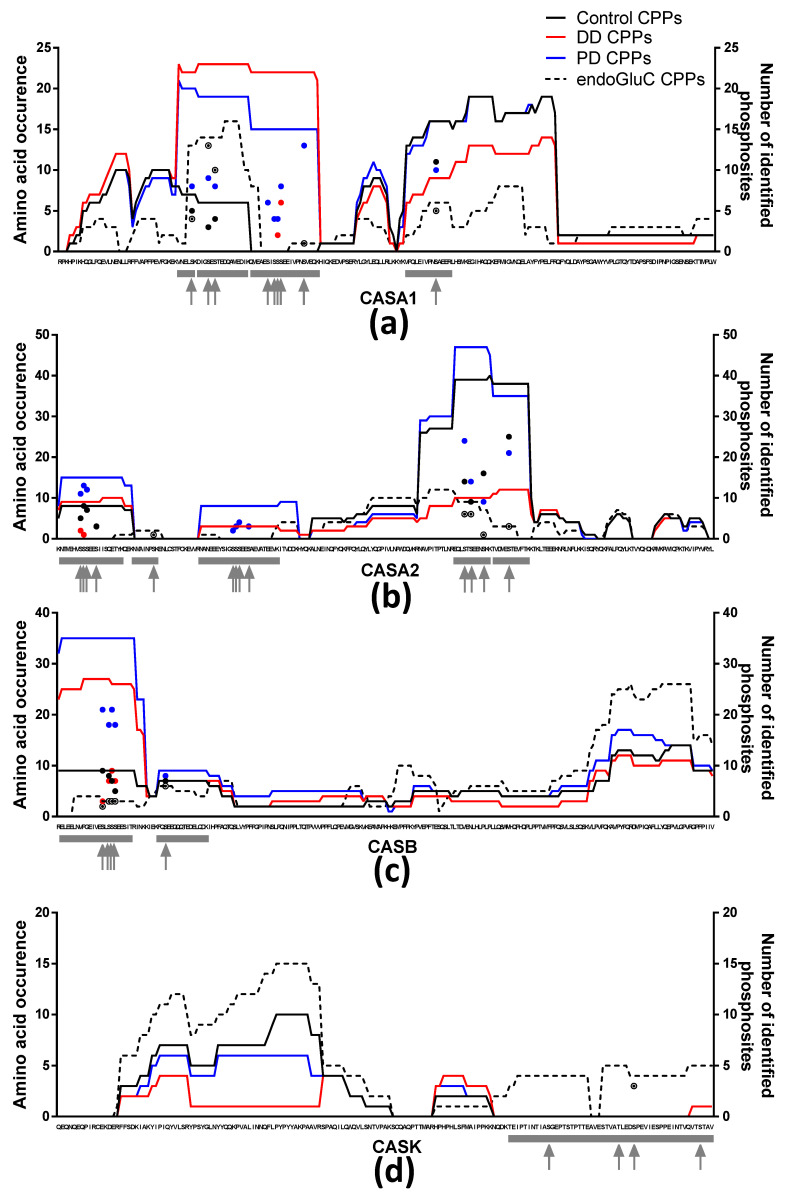
Peptide patterns of the control- (black lines and black dots), DD- (red lines and red dots), PD- (blue lines and blue dots) and endoGluC- (black dashed lines and black dashed dots) CPP hydrolysates for (**a**) alpha-S1 casein (CASA1), (**b**) alpha-S2 casein (CASA2), (**c**) beta-casein (CASB) and (**d**) kappa-casein (CASK). The *X*-axis represents the amino acid sequence of the different proteins, and the left *Y*-axis represents amino acid occurrence, whereas the right *Y*-axis represents the number (colored dots) of phosphosites identified for each hydrolysate. Gray rectangles represent in-silico-predicted tryptic CPPs (without missed cleavage) and gray arrows represent phosphorylation sites referenced in UniprotKB/Swiss-Prot (1 June 2021).

## Data Availability

Publicly available datasets were analyzed in this study. The mass spectrometry proteomics data have been deposited to the ProteomeXchange Consortium via the PRIDE [24] partner repository with the dataset identifier PXD027132.

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
