# Peer review of "Partial-, Double-Enzymatic Dephosphorylation and EndoGluC Hydrolysis as an Original Approach to Enhancing Identification of Casein Phosphopeptides (CPPs) by Mass Spectrometry"

_foods, 2021, doi:10.3390/foods10092134_

Round 1

Reviewer 1 Report

The article “Partial-, double enzymatic-dephosphorylation and endoGluC hydrolysis as an original approach to enhancing identification of casein phosphopeptides (CPPs) by mass spectrometry” proposes the use of a double- and a partial-enzymatic de-phosphorylation of CPPs as well as an enzymatic hydrolysis using an endoGluC protease before peptidomics analysis.

The work is well-written and data clearly organized and presented with nice graphics.

However, I recommend that the authors point out that, through the proposed methodological approach, is not possible to map new phosphorylation sites. Indeed, the proposed experimental workflow is only useful to compare the known phosphorylation sites of a protein with peptides from the same protein obtained by the 4-phase strategy, including de-phosphorylation ad double enzymatic digestion.

Following, some minor points to address to improve your manuscript:

- I suggest to use a schematic representation of the: samples preparation, MS-analyses and data comparison to better help the reader in understanding the goal of the proposed approach.

- In Figure 1, please, better clarify the correspondence of the lowercase letters with the samples compared in the 2-way ANOVA test. A legend could be useful.

- Please, apply the proper correction at the lines 214, 228, 253, 270, 278, 294, 303 that report the sentence “Error! Reference source not found”.

- A full stop is missing at line 107.

- Pleas, try to increase the letter size within Figure 2. Probably, increasing the size of the whole figure could be useful.

- Please, define the abbreviations “CASA, CASB and CASK” also in the main text and not only in the figure legend.

Reviewer 2 Report

This manuscript presents a very interesting study on an important aspect of peptidomics/proteomics in the food /dairy industry.

The study is well planned and executed, the manuscript well written, even if somewhat spoiled by numerous if small English languages issues. I have tried to highlight most of them in the attached document, but I would encourage the authors to get a colleague with good English to get over their manuscript for a final proofread.

One last minor issue (that can be resolved at editorial stage) is the conflicted links to figures within the text returning errors, and the fact that figures 2 (a to d) could be presented in a larger format for an easier read, as they contain most if not all of the information upon which the discussion is based.   
